# Effect of the Surface Hydrophobicity Degree on the *In Vitro* Release of Polar and Non-Polar Drugs from Polyelectrolyte Matrix Tablets

**DOI:** 10.3390/polym10121313

**Published:** 2018-11-27

**Authors:** Cristhian J. Yarce, Juan D. Echeverri, Constain H. Salamanca

**Affiliations:** Programa de Maestría en Formulación de Productos Químicos y Derivados, Facultad de Ciencias Naturales, Universidad Icesi, Calle 18 No. 122-135, Cali 76003, Colombia; cjyarce@icesi.edu.co (C.J.Y.); jdecheverri@icesi.edu.co (J.D.E.)

**Keywords:** surface properties, drug release, carbamazepine, metoprolol succinate, polyelectrolyte materials

## Abstract

This work is the continuation of a series of studies focused on establishing the relationship between the surface thermodynamic properties of polyelectrolyte matrix tablets and drug release mechanisms. In this case, two model drugs with different polarity features, such as carbamazepine (non-polar) and metoprolol succinate (polar) were used in combination with polymeric material hydroxypropyl-methyl cellulose (HPMC) and two polyelectrolytes derived from maleic anhydride corresponding to the sodium salts of poly(maleic acid-*alt*-ethylene) and poly(maleic acid-*alt*-octadecene) named PAM-0Na and PAM-18Na, respectively. The polymers were obtained and characterized as reported previously. Surface studies were performed by the sessile drop method, whilst the surface free energy was determined through Owens, Wendt, Rable and Kaeble (OWRK) semi-empirical model. By contrast, the drug release studies were performed by in vitro dissolution tests, where data were analyzed through dissolution efficiency. The results showed that, depending on the drug polarity, type and polymer proportion, surface properties and drug release processes are significantly affected.

## 1. Introduction

The polymers derived from maleic anhydride correspond to a class of materials that have shown an interesting potential as pharmaceutical excipients due to their biocompatibility and diversity of applications in product formulation [1]. Some of these materials are the salts of poly(maleic acid-*alt*-octadecene), which have described several interesting properties, such as (i) the capability to form hydrophobic “pseudo-phases” in aqueous media, useful for the inclusion of drugs, [2] and (ii) the capability to generate surface with different hydrophobicity degrees, useful in the development of modified-release matrix systems [3]. The latter has been the subject of a study carried out in our research laboratory, where we have focused on establishing the relationships between the thermodynamic properties of matrix tablets surface and the mechanisms of the in vitro release of active pharmacological ingredients. The first stage of the study was focused on establishing the effect of the potassium salt of poly(maleic acid-*alt*-octadecene) on (i) the surface of various tablets of binary composition and (ii) the release of the model polar drug (ampicillin trihydrate) [4]. The second stage was focused on establishing the effect of the polymeric hydrophobicity degree on the tablet surfaces and the kinetic release of the same model drug [5]. The third and current research are focused on determining the effect of the polarity of the drug released, with respect to the surface hydrophobicity degree and the proportion of polymeric material in the binary matrix tablets. For this, two model drugs and two polymers with different polarity and hydrophobicity degrees were used, respectively. Such drugs were carbamazepine CBZ (non-polar) and metoprolol succinate MTP (polar) named and, respectively, whilst the polymers were the sodium salts of poly(maleic acid-*alt*-methylene), hydrophilic polymer, and poly(maleic acid-*alt*-octadecene), amphiphilic polymer, referred as PAM-0Na and PAM-18Na, respectively. In addition to polymers derived from maleic anhydride, the reference material hydroxypropyl methylcellulose-HPMC was used. The chemical structures of the study materials are shown in Figure 1.

At last, the authors intend with this work to establish a relationship between the surface thermodynamic properties in matrix tablets with respect to the release of drugs with different polarity characteristics.

## 2. Materials and Methods

### 2.1. Materials

The model drugs were CBZ (Fersinsa Gb) and MTP, both provided by Tecnoquímicas S.A laboratories (Cali, Colombia), which were used as received. The polymeric precursor materials were poly(maleic anhydride-*alt*-ethylene)-PAM-0 with *M*_W_ ~100,000 (Sigma-Aldrich, San Louis, MO, USA), poly(maleic anhydride-*alt*-octadecene)-PAM-18 with *M*_W_ of 30,000–50,000 (Sigma-Aldrich, San Louis, MO, USA) and hydroxypropyl methylcellulose (HPMC) with *M*_W_ of 86,000 (Sigma-Aldrich, San Louis, MO, USA). The reagents used for the preparation of the dissolution media were KOH, KCl, KH_2_PO_4_, K_2_HPO_4_ and water type I (Arium pro Sartorius Stedim biotechnology VF, Göttingen, Germany). For contact angle measurements, reference liquids were used: ultra-pure water type I, isopropanol (LiChrosolv, Merck KGaA, Darmstadt, Germany) and ethylene glycol (Merck KGaA, Darmstadt, Germany). Solutions with pH values of 1.2 and 7.4 with the ionic strength of 0.15M were prepared from mixtures of HCl/KCl and KH_2_PO_4_/K_2_HPO_4_ solutions, respectively. KCl was used to adjust the ionic strength. The criteria for the preparation of the buffer solutions were taken from modifications of those stated in the current pharmacopeia (USP 40 – NF 35) [6].

#### Collection and Characterization of the Polymeric Materials

PAM-0Na and PAM-18Na were obtained as previously described [5]. Briefly, 100 g of precursor polymeric materials PAM-0 and PAM-18 were hydrolyzed separately in 2 L of ultra-pure pure water mixed with NaOH in a 1:1 molar ratio (according to the co-monomeric unit of the precursor materials PAM-0 and PAM-18). The modification was carried out at room temperature for 24 h under moderate agitation (200 rpm). Subsequently, each polymer solution was dialyzed using cellulose membrane (12 kD cut-off size) and pre-concentrated through a stirred ultrafiltration cell (Amicon^®^ cells 8400, Merk-Millipore, Billerica, MA, USA) with a 12 kDa cut-off polyethersulfone (PES) membrane. Afterwards, each polymer solution was lyophilized (model FDU 1110, Eyela, Tokyo Rikakikai, Tokyo, Japan) until obtaining solid materials with a yield higher than 90%, which was sieved through 75 μm mesh (number 200). On the contrary, the structural characterization of polymeric materials PAM-0Na and PAM-18Na was performed in an infrared spectrometer (Nicolet 6700, Thermo Fisher Scientific, Waltham, MA, USA), where spectral signals of the precursor materials, PAM-0 and PAM-18, were compared.

### 2.2. Methods

#### 2.2.1. Characterization of Powder Polymeric Materials

Two powder parameters were used, such as the angle of repose and the Carr and Hausner indexes [7,8]. These parameters show the capability of powder polymeric materials to flow. The flowability degree was obtained through a powder flow tester (Erweka GmbH), whilst the percentage of compressibility was determined using a density meter (Logan Tap-2S, Logan Instrument, Somerset, NJ, USA) [9]. Briefly, 50 g of polymeric materials were dried at 100 °C until reaching a constant weight.

#### 2.2.2. Thermal Characterization of Polymer-Drug Blends

The model drugs, PAM-0Na and PAM-18Na, and their respective mixtures in proportions of 10%, 20%, 30% and 40% (*w*/*w*) were analyzed using a Q2000 differential scanning calorimeter (DSC; TA Instruments, New Castle, DE, USA) calibrated with indium *T*_m_ = 155.78 °C and ∆*H*_m_ = 28.71 J/g. DSC analysis was carried out using three heating cycles from −90 °C (183.15 K) to 200 °C (523.15 K) with a heating rate of 20 °C/min.

#### 2.2.3. Preparation of Tablets

Each binary tablet was elaborated using 200 mg of CBZ and 100 mg of MTP with PAM-0Na, PAM18Na and HPMC polymers by random blending process. Different polymer proportions corresponding to 0%, 10%, 20%, 30% and 40% *w*/*w* were used. The tableting process was performed using a homemade tableting machine with ¼ inch stainless steel flat punches. A compression pressure of 300 Psi was applied for 10 s in each tableting. The diameter for each tablet was 13 mm and the surface area for tablets was around 82 mm^2.^ The hardness was determined using a durometer (Logan HDT-400, Somerset, NJ, USA), whilst the disintegration time was determined by an automated disintegrator (Logan USP DST-3, Somerset, NJ, USA) in type II water at 37 °C.

#### 2.2.4. Contact Angle Measurements

The determination of the static contact angle was performed on the surface of each tablet of CBZ and MTP alone and with the polymeric materials (PAM-0Na, PAM-18Na and HPMC) using different polymer proportions, immediately after the manufacture of the tablets. For each system, the sessile drop method was used using a contact angle meter (OCA15EC Dataphysics Instruments, Filderstadt, Germany) with the software SCA20 version 4.5.14. The experimental conditions have been widely described in previous studies regarding the degree of hydrophobicity of different polymeric materials [3].

#### 2.2.5. Determination Thermodynamics Surface Parameters

The surface free energy (SFE) was determined from the Owens, Wendt, Rable and Kaeble model (OWRK) [10], where three liquids of increasing polarity were used, namely, isopropanol, ethylene glycol and water. The OWRK model allows obtaining the total surface free energy (SFE_Total_) and discriminates the type and degree of interaction that occur between the solid surfaces with the liquid drop. In this model, it is possible to differentiate the SFE of the solid (*γ_SV_*) with respect to two kinds of interactions, namely, dispersive type (van der Waals interactions) and polar type (dipole-dipole interactions and hydrogen bonds). The OWRK model is defined by a linear equation y = mx + b, as follows:(1)$$\frac{\gamma_{LV}(\cos\theta +1)}{2{\left(\gamma_{LV}^{D}\right)}^{1/2}}={\left(\gamma_{SV}^{P}\right)}^{1/2}\frac{{\left(\gamma_{LV}^{P}\right)}^{1/2}}{{\left(\gamma_{LV}^{D}\right)}^{1/2}}+{\left(\gamma_{SV}^{D}\right)}^{1/2}\;$$
$$\mathrm{where}:\;Y=\frac{\gamma_{LV}(\cos\theta +1)}{2{\left(\gamma_{LV}^{d}\right)}^{1/2}},\;m={\left(\gamma_{SV}^{p}\right)}^{1/2},\;X=\frac{{\left(\gamma_{LV}^{p}\right)}^{1/2}}{{\left(\gamma_{LV}^{d}\right)}^{1/2}},\;b={\left(\gamma_{SV}^{d}\right)}^{1/2}\;$$

In this equation, *p* and *d* correspond to the dispersive and polar contributions, respectively, whilst *θc*, *γ_SV_* and *γ_LV_* are the contact angle, surface tension of the solid and surface tension of the liquid, respectively.

#### 2.2.6. In Vitro Dissolution Test

The dissolution studies were carried out using the paddle method in a previously calibrated dissolver (apparatus II, Vision G2 Classic 6-Hanson, Chatsworth, CA, USA). The speed of the paddle was 100 rpm at 37 ± 0.5 °C (310.15 K). The volume of the simulation media for gastric and plasma conditions (buffer solutions pH 1.2 and pH 7.4 with the ionic strength of 1.5 M, respectively) was 900 mL. Each dissolution test was performed for 8 h (480 min), where 5 mL of the sample was taken, with media replacement at predetermined time intervals. The samples taken were passed through 0.45-m filters. The amount of CBZ and MTP was determined by UV spectrophotometry, using a Microplate reader, Synergy H1 (Biotek Instruments Inc., Winooski, VT, USA), with previously standardized methods. For this purpose, two wavelengths of 285 and 275 nm were used for CBZ and MTP, respectively. Specificity, linearity and precision were determined concerning repetitiveness and non-exhaustive reproducibility. Data obtained from in vitro dissolution profiles are reported as the average dissolution efficiency (ED) of the tablets. This parameter is defined as the area under the dissolution curve (AUC) recorded at a given time in relation to the rectangular area (*R*) described by 100% of the solution at the same time. The dissolution efficiency can be calculated from:(2)$$D.E=\frac{AUC}{R}\times 100\%=\frac{\int_{0}^{t}y\times dt}{y_{100}\times t}\times 100\%\;$$
where *y* is the percentage of drug dissolved at time *t*, *y*_100_ is the percentage of drug dissolved, assuming 100% release.

#### 2.2.7. Processing and Data Analysis

All data were tabulated and analyzed using Microsoft Excel and GraphPad Prism 6 (GraphPad Software, La Jolla, CA, USA), where a level of confidence of 95% was adopted, and data were expressed as the mean ± standard deviation.

## 3. Results and Discussion

### 3.1. Collection and Characterization of the Polymeric Materials

The formation of PAM-0Na and PAM-18Na polymers was evidenced by a physical change. The solution passes from a heterogeneous mixture to a completely homogeneous solution, due to the opening of the maleic group in PAM-0 and PAM-18 polymers, which produces carboxylic acid and carboxylate groups in the polymer backbone. The structural changes of the PAM-0 and PAM-18 precursors and the PAM-0Na and PAM-18Na polymer derivatives were analyzed by comparing the Fourier transform infrared (FT-IR) spectra. Briefly, the FT-IR spectra of the polymers derived from PAM-0 and PAM-18 showed typical signals corresponding to the symmetric and asymmetric stretching values of the alkyl chain between ~2920–2940 and ~2850–2875 cm^−1^, respectively. Likewise, common signals of the hydroxyl group from carboxylic acid between ~3200 and 3600 cm^−1^ were observed, corroborating the presence of carboxylic acid and carboxylate forms in PAM-0Na and PAM-18Na, respectively. Finally, the signals corresponding to different forms of the carbonyl group, such as ~1856–1778 cm^−1^ (maleic anhydride of PAM-0), ~1664–1560 cm^−1^ (carboxylates of PAM-0Na), ~1778–1708 cm^−1^ (maleic anhydride of PAM-18) and ~1645–1564 cm^−1^ (carboxylates of PAM-18Na), were observed.

### 3.2. Characterization of Powder Polymeric Materials

The results of the particle study are summarized in Table 1, according to the guidelines established in the USP 40/NF35 [6].

The Carr index (% compressibility) with values ≤10% and between 11% and 25% suggest excellent and passable-poor flow properties, respectively, whilst the Hausner ratio with values between 1.0–1.1and 1.2–1.3 indicate excellent and passable-to-poor flowability in the powder material. On the contrary, the angles of repose values from 25–30 and 30–40 suggest excellent and good-to-intermediate flowability properties, respectively. Therefore, our results show that all the study materials presented intermediate-to-poor flow properties, except for PAM-0Na, which demonstrated excellent flowability properties. This result is consistent with the drying method (lyophilization), where polymer plates are collected and subsequently sifted.

### 3.3. Thermal Characterization of Polymer-Drug Blends

The thermograms of CBZ and MTP drugs and the PAM-18Na, PAM-0Na and HPMC polymers, as well as their respective mixtures, are shown in Figure 2. 

The DSC thermograms of the first heating cycle for the polymeric materials PAM-0Na, PAM-18Na and HPMC showed a thermal event between ~150 and 170 °C which is attributed to the presence of water bound to the polymeric materials [11,12,13,14,15]. On the contrary, CBZ showed an endothermic signal at ~180 °C, corresponding to the fusion of polymorph III which was already characterized by others authors [16,17], whilst the MT showed an endothermic signal ~140 °C attributed to the melting of polymorph I for its structure [18]. In the case of CBZ with PAM-0Na, a thermal event was observed, which increased with the rise of the polymer amount in the mixture. Such new signals from thermophysical transitions could indicate a dynamic change in spatial distributions between the drug and the polymer similar to the formation of molecular adducts or clathrates [19,20,21,22,23]. In the case of MTP with PAM-0Na, the interactions between the components did not show formation of a new signal. However, a slight displacement in the melting signal of the drug was noted. In case of CBZ and PAM-18Na polymer, shift in the thermophysical signal of the pure components was observed, which could be caused by the slight electrostatic repulsion between the mixing components. This same behavior was also appreciated with the MTP drug. Finally, the combination of CBZ and MTP with the HPMC polymer showed a displacement of the thermophysical transition signals, leading to a combination of signals that revealed new spatial rearrangements of the materials in such a state of mixing [24,25,26].

### 3.4. Physical Characterization of the Binary Tablets

The results of the hardness and disintegration time for the CBZ and MTP tablets elaborated with different proportions of polymeric materials (PAM-0Na, PAM-18Na and HPMC) are summarized in Table 2. First, it is important to highlight the low values of hardness reached in the most processed tablets. However, despite this condition, it was possible to achieve extended disintegration times, mainly in CBZ tablets with polymer percentages of 30% and 40%. In the case of CBZ with PAM-0Na and HPMC polymer, there was an increase in tablet hardness with respect to polymer concentration. By contrast, with the PAM-18Na polymer, the tablet hardness remained constant with values close to those displayed by CBZ alone. Moreover, a very interesting result was observed on the disintegration time of CBZ tablets, that is, the CBZ tablets alone showed a longer disintegration time than the tablets mixed with the PAM-0 and PAM-18 polymers. This result suggests that such polymers could affect the erosion process in the tablet; whereas with the HPMC polymer, by contrast, the disintegration times were greater than 4 h because this material forms gelled systems that do not disintegrate quickly [27]. In contrast, the MTP tablets presented lower results in tablet hardness than those shown by CBZ tablets. Such result can be explained by the low agglutination capability of both the MTP and polymeric in the dry mix. In comparison, the results of the disintegration time showed an increase with the increment of the polymer amount in the tablet.

Regarding the MTP tablets with the PAM-0Na polymer, the disintegration time passed from 14 to 33 min, describing the typical behavior of a conventional tablet. This result makes sense if both the drug and the polymer have affinity for the aqueous media; thus, the disintegration process is faster. Conversely, with the PAM-18Na polymer, there was an increase in the disintegration time that could be explained by the higher degree of hydrophobicity given by the polymeric alkyl chain; thus, the interaction with the media is less favored than with the polymer PAM-0Na. With respect to MTP a tablet with HPMC, a similar behavior was observed with the PAM-18Na where only a proportion of 40% displayed the longest disintegration time. This can be explained by the gelation effect, which has been widely attributed to such polymer.

### 3.5. Contact Angle Measurements

The interaction effect given between ultra-pure water and the tablet surfaces formed with the CBZ and MTP drugs and the polymeric materials PAM-0Na, PAM-18Na and HPMC are shown in Figure 3. Results showed that the surface of CBZ tablets without polymer showed a contact angle value (θ_c_) of 100.5°, suggesting hydrophobic features. On the contrary, when the polymeric material is incorporated inside the tablets, a change in θc occurs, and depending on the type and amount of the polymer, such effects are more obvious. Moreover, even with small polymer amount incorporated (10%), the tablet surfaces shifted from the “non-wettability” zone (θ_c_ > 90°) to the ‘wettability’ zone (θ_c_ < 90°). With regard to the CBZ tablets with the PAM-0Na polymer, θ_c_ decreases gradually with the increase of the polymer until reaching a value of 62.2°, indicating that the tablet surface becomes less hydrophobic. In contrast, the PAM-18Na and HPMC polymer materials showed that as the polymer amount increases in the tablet, the θc values also increase from 82° to 99° and 81° to 93°, respectively, enhancing the hydrophobicity degree of the tablet surface. On the contrary, the tablet surface formed only by MTP showed hydrophilic features (θ_c_ = 56.5°), becoming slightly less hydrophilic with the incorporation of PMA-18Na and HPMC polymers inside the tablet. However, with the polymer PAM-0Na, a different behavior was observed, which did not depend on the amount of polymer used.

### 3.6. Determination Thermodynamics Surface Parameters

The results of the thermodynamic assays evaluated by the semi-empirical model OWRK for the tablet surface of CBZ and MTP and the polymeric materials PAM-0Na, PAM-18Na and HPMC are shown in Figure 4. 

The results showed that CBZ and MTP tablets with the PAM-0Na, PAM-18Na and HPMC polymeric materials exhibited different behaviors depending on the type and polymer amount incorporated in the tablets. In the case of the CBZ tablets without polymer, the contribution to the SFE was mainly given by dispersive interactions (van der Waals). Whilst with the hydrophilic polymer PAM-0Na, the SFE increased with the polymer amount, where the dispersive contribution decreased, and the polar contribution (hydrogen bond) increased, inverting the hydrophobic character of the tablet. In contrast, the CBZ tablets with PAM-18Na and HPMC showed that at polymer proportions of 10% and 20%, the contribution to the SFE was given by both dispersive and polar interactions in a similar way, whereas at polymer proportions of 30% and 40%, the contribution to the SFE was mainly given by the dispersive contribution, and this effect is greater with PAM-18Na than HPMC. For MTP and PAM-0Na tablets, no relationship was observed between the polymeric amount and SFE values. However, it was possible to notice that the polar contribution was always the greatest. On the contrary, the PAM-18Na and HPMC polymers showed that their incorporation inside the tablets led to a decrease in SFE. In both cases, a marked tendency was observed, where the dispersive contribution increased, and the polar contribution decreased with the polymer amount, and this effect is greater with PAM-18Na than HPMC, whilst at polymer proportions of 30% and 40%, the contribution to the SFE was mainly given by the dispersive contribution, and this effect is greater with PAM-18Na than HPMC. For MTP and PAM-0Na tablets, no relationship was observed between the polymeric amount and SFE values. However, the polar contribution was always the most significant contribution. On the contrary, the PAM-18Na and HPMC polymers showed that their incorporation inside the tablets led to a decrease in SFE. In both cases, a marked tendency was observed, where the dispersive contribution increased, and the polar contribution decreased with the polymer amount, and this effect is greater with PAM-18Na than HPMC. These results are very interesting, as they suggest that polymers can be found in very specific forms on the tablets surfaces, interacting differently with liquids of interest such as biological fluids. In this way, the modulation in the surface hydrophobicity degree could modulate the manner in which the drug is released from a matrix system, as discussed below.

### 3.7. In Vitro Dissolution of CBZ and MTP

The results of the in vitro dissolution profiles for each tablet of CBZ and MTP elaborated with PAM-0Na, PAM-18Na and HPMC materials at different polymer ratios, in two physiological simulation media such as gastric medium (buffer solution pH: 1.2, 0.15 M) and duodenal (solution buffer pH: 7.4, 0.15 M), are presented in Figure 5, whilst the results of dissolution efficiency are summarized in Table 3.

The results showed a marked dependence of the release profiles with respect to the (i) dissolution media, (ii) type of drug and (iii) polymer proportion within the tablet. In the case of CBZ tablets without polymer, the percentage of dissolution efficiency was less than 10% in both media. However, when the PAM-0Na polymeric materials were incorporated in the tablet, an increase in the percentage of dissolution efficiency was observed, which achieved values higher than 60% and 30% in duodenal and gastric media, respectively. This result may be explained considering several aspects, such as a rapid and complete solubilization of the polymer in both media and the formation of pH-dependent interpolymeric aggregates [28]. At gastric pH, there are no charges on the PAM-0Na polymeric surface leading to the formation of polymeric networks that improve the solubilization of CBZ. Whilst at duodenal pH, the polymer PAM-0Na acquires a fraction of charge that extends the polymer and prevents the polymer-polymer aggregation affecting the solubilization of CZB. Regarding to the PAM-18Na and HPMC polymers, a similar tendency is observed, where the increase of the polymer amount in the tablets leads to a decrease in the drug released toward the media. This behavior could be explained considering that a greater amount of these polymers in the tablets, the disintegration time in tablet increase affecting the drug release. In the case of MTP tablets, the type of polymer and the media pH did not significantly affect the dug release profiles and the dissolution efficiency, showing typical profiles of conventional dosage forms. The most significant modulation effects occur with the PAM-18Na, where the percentages of metoprolol released tend to decrease as the proportion of the polymeric material in the system increases. This effect could be explained considering that the increase in the polymer amount leads to rise on the tablet disintegration time, disturbing the MTP release towards the medium. In addition, such PAM-18Na polymer seems to have a competition for the solvation with the media, affecting, in turn, the dissolution of MTP.

At last, it is important to mention that the kinetic analysis of the release of the drugs was also carried out with respect to different semi-empirical models in order to establish the drug release mechanism. Nevertheless, the fits to the models used did not show (in some cases) an adequate adjustment to be able to attribute a specific mechanism. On the contrary, the kinetic analysis showed there are several drug release mechanisms involved as erosion and swelling of the matrix. The results are presented in the Appendix A.

## 4. Conclusions

The physical, superficial and kinetic characteristics of the matrix tablets of CBZ and MTP are strongly affected by the type and proportion of the polymer employed, which can generate modifications in the physical features and drug release mechanism. In this way, hydrophilic polymers do not interact adequately with non-polar drugs like CBZ, generating tablets with a porous or rough physical structure, whilst with MTP the polymers could generate tablets of smooth surfaces and better cohesiveness. In terms of the surface properties, the PAM-0Na polymer decreases the contact angles and increases the SFE of tablets, regardless of the polarity of the drug, suggesting that the use of this polymeric material could be an interesting strategy to enhance the interaction between aqueous media and poorly soluble drugs, such as CBZ. Respect to PAM-18Na polymer, this behaves as an ‘intelligent’ material that depends on the proportion used in the tablet, and the pH of the medium can have different release mechanisms. For example, at polymer ratios of 10–20%, it can act as a solubility enhancer, regardless of the media pH. However, at polymer ratios of 30–40%, it can perform similar to HPMC, where it generates a gel layer that affects the diffusion of drugs from the tablet. The use of PAM-18Na and PAM-0Na polymers can (i) improve the solubility of non-polar drugs or (ii) delay the release of polar drugs in aqueous media. In the case of the PAM-0Na polymer, it is possible to improve the solubility of non-polar drugs such as CBZ in different types of media, whereas with the polymer PAM-18Na, the release of polar and non-polar drugs can be controlled, depending on the polymer proportion used in the tablet. These polymers thus reaffirm their potential application as controlled release excipients for drugs with different polarity features.

## Figures and Tables

**Figure 1 polymers-10-01313-f001:**
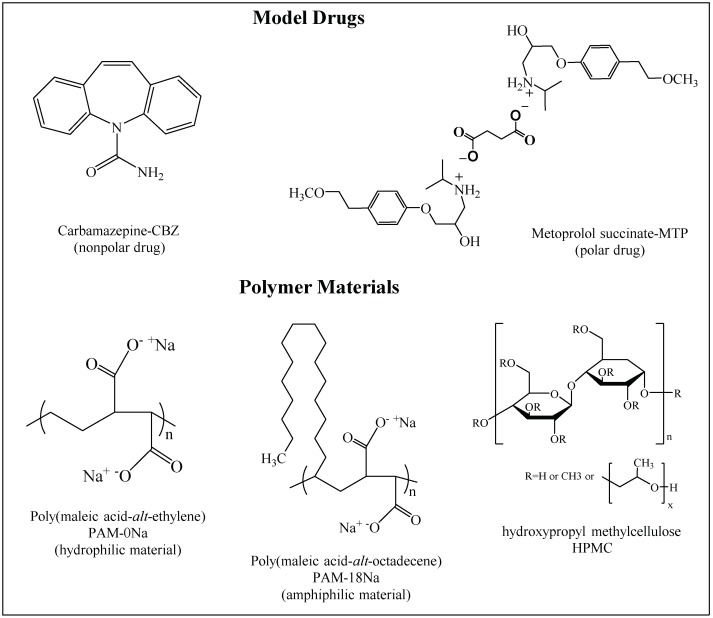
Chemical structure of the Carbamazepine and Metoprolol (model drugs), the polyelectrolyte materials derived from maleic anhydride and the reference polymer hydroxypropyl-methyl cellulose (HPMC).

**Figure 2 polymers-10-01313-f002:**
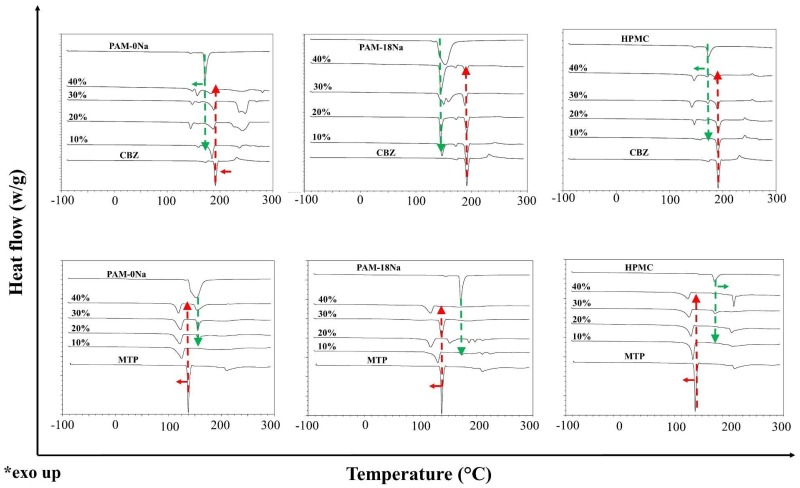
DSC thermogram of mixtures of carbamazepine (upper CBZ) and metoprolol (lower MTP) with the PAM-18Na; PAM-0Na; HPMC polymers at 10%, 20%, 30% and 40% HPMC, hydroxypropyl-methyl cellulose.

**Figure 3 polymers-10-01313-f003:**
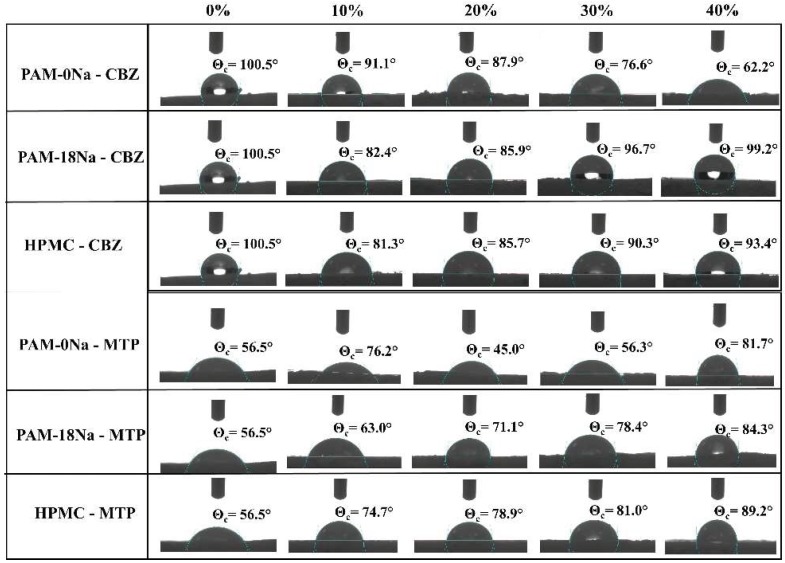
Contact angle values for tablets elaborated with CBZ and MTP in different proportions of PAM-0Na, PAM-18Na and HPMC polymers.

**Figure 4 polymers-10-01313-f004:**
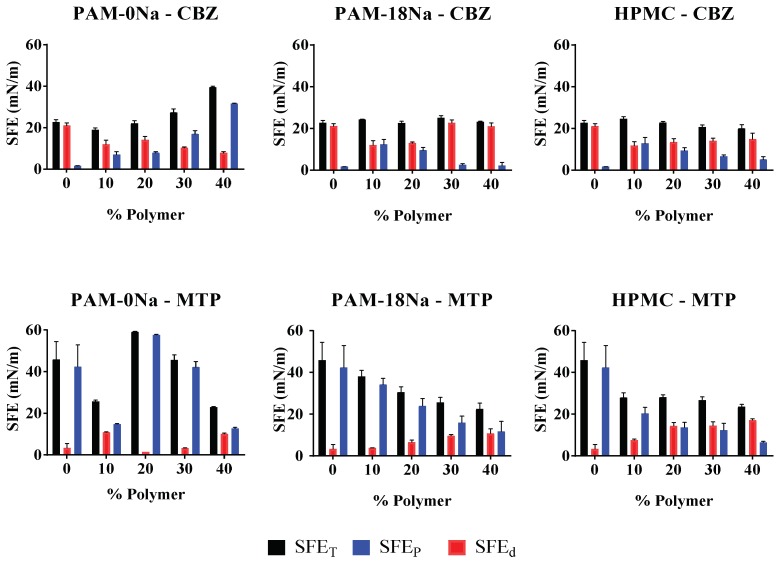
Surface Free Energy-SFE for tablets elaborated with CBZ and MTP and different proportions of the PAM-0Na, PAM-18Na and HPMC polymers.

**Figure 5 polymers-10-01313-f005:**
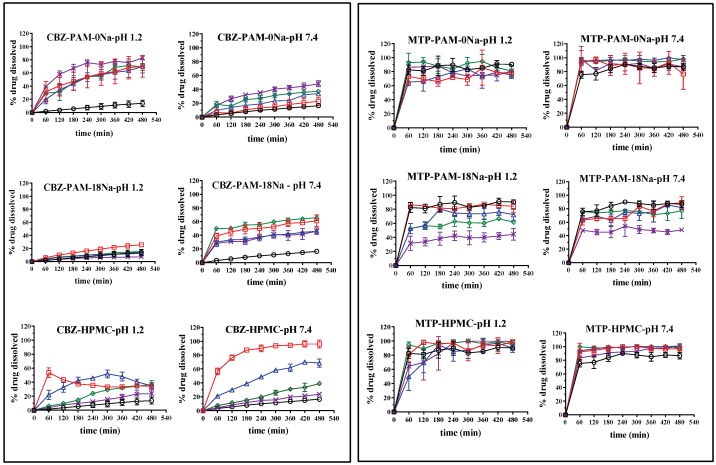
In vitro dissolution profiles of carbamazepine (CBZ) and metoprolol succinate (MTP) tablets elaborated with different polymer ratios: ◯ = 0%, 
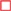
 = 10%, 
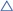
 = 20%, 
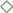
 = 30% and 

 = 40% and two media of physiological simulation.

**Table 1 polymers-10-01313-t001:** Powder properties of the study materials.

Parameter	PAM-0Na	PAM-18Na	HPMC	CBZ	MTP
Bulk density g/mL	0.54 ± 0.02	0.24 ± 0.02	0.36 ± 0.01	0.42 ± 0.02	0.46 ± 0.01
Tapped Density g/mL	0.60 ± 0.01	0.28 ± 0.15	0.47 ± 0.01	0.52 ± 0.01	0.50 ± 0.01
Carr Index (%)	10.10 ± 2.3	12.7 ± 1.2	21.6 ± 2.0	18.5 ± 0.8	7.2 ± 2.0
Hausner Index (%)	1.10 ± 0.02	1.2 ± 0.01	1.3 ± 0.03	1.2 ± 0.01	1.1 ± 0.02
Angle of repose (°)	23.50 ± 2.8	36.3 ± 2.3	26.9 ± 1.4	27.7 ± 2.0	40.4 ± 2.2

**Table 2 polymers-10-01313-t002:** Results of hardness and disintegration time for tablets elaborated with CBZ and MTP and different proportions of the PAM-0Na, PAM-18Na and HPMC polymers.

Drug	Polymer Material	% Polymer	Hardness (kp)	Disintegration Time (min: s ± s)
Carbamazepine (CBZ)		0	2.48 ± 0.44	>4 h
	PAM-0Na	10	1.41 ± 0.29	24:40 ± 1.53
		20	2.11 ± 0.69	37:00 ± 1.73
		30	4.29 ± 0.34	44:30 ± 0.58
		40	8.82 ± 0.31	65:40 ± 1.15
	PAM-18Na	10	1.56 ± 0.73	28:40 ± 1.53
		20	1.55 ± 0.45	47:30 ± 2.52
		30	1.62 ± 0.11	56:10 ± 1.03
		40	1.89 ± 0.05	67:30 ± 2.52
	HPMC	10	3.59 ± 1.01	>4 h
		20	5.92 ± 0.97	>4 h
		30	5.81 ± 1.58	>4 h
		40	8.24 ± 0.90	>4 h
Metoprolol succinate (MTP)		0	1.97 ± 0.37	16:40 ± 1.52
	PAM-0Na	10	1.56 ± 0.15	14:20 ± 1.15
		20	<0.60	19:40 ± 1.53
		30	<0.60	24:40 ± 0.58
		40	<0.60	29:30 ± 1.15
	PAM-18Na	10	0.84 ± 0.28	33:40 ± 0.58
		20	0.59 ± 0.14	54:30 ± 1.15
		30	0.64 ± 0.06	54:40 ± 0.58
		40	0.64 ± 0.31	59:40 ± 1.53
	HPMC	10	1.26 ± 0.07	19:20 ± 1.15
		20	0.93 ± 0.13	24:40 ± 0.58
		30	0.67 ± 0.16	64:40 ± 1.53
		40	1.48 ± 0.98	106:20 ± 1.15

**Table 3 polymers-10-01313-t003:** Values of dissolution efficiency (*DE*) for CBZ and MTP tablets at different proportions of polymeric materials at 37°C, using two media of physiological simulation.

Drug	Media	% Polymer	Dissolution Efficiency Percentage (%)		
			PAM-0Na	PAM-18Na	HPMC
CBZ	Gastric	0	7.5	7.5	7.5
		10	49.8	15.6	37.1
		20	46.1	8.5	38.1
		30	48.7	9.9	21.1
		40	63.3	5.1	13.0
	Duodenal	0	9.4	9.4	9.4
		10	12.7	48.9	81.3
		20	19.6	36.6	45.6
		30	25.5	55.4	20.6
		40	32.6	34.9	13.4
MTP	Gastric	0	86.0	86.0	86.0
		10	72.8	80.7	93.1
		20	73.5	68.1	84.2
		30	88.8	57.5	95.5
		40	79.1	37.5	90.0
	Duodenal	0	83.4	83.4	83.4
		10	88.7	70.2	99.4
		20	96.9	71.7	102.0
		30	94.2	71.5	97.7
		40	87.7	46.3	89.0

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
