# Peer review of "Effect of the Surface Hydrophobicity Degree on the In Vitro Release of Polar and Non-Polar Drugs from Polyelectrolyte Matrix Tablets"

_polymers, 2018, doi:10.3390/polym10121313_

Reviewer 1 Report

The authors examine the release kinetics of two model drugs with different polarities from two polymer formulation of hydroxypropyl-methyl cellulose/maeleic anhydride derivatives -PAM-0Na and PAM-18Na. A polar drug, metroprolol succinate and non-polar drug carbamazepine were released via dissolution in vitro and characterized. The polymer formulation were characterized and the ability to enable the release of the model drugs at two different physiologically relevant conditions. The study appears to be one in a series of studies by the authors to examine drug delivery formulations and drug release kinetics of model drug (s).

Overall, the paper is well-presented and the characterization methods followed are relevant to the topic. Below are my comments.

Structure of HPMC needs to be added to the chemical structures of materials used in the paper. The number of samples analyzed for each tests needs to be stated.

Can the authors discuss the effect of surface porosity and inhomogeneity on the contact angle measurements in the polymer-drug formulation in Figure 3?

Can the authors comment on the consistency and repeatability of the characterization methods (FTIR) used to formulate the PAM-0Na and the PAM-18Na polymers?

Accuracy check over-all- assuming the authors meant PAM-18Na in line 174 and the phrase in line 78-

The X and Y axis in graphs in Figure 2.  needs to be adjusted to bigger font size for clarity.

As the authors have briefly mentioned,  release from polyelectrolyte-based polymeric formulation with added differences in polarity in the encased drug is dependent on multiple factors and a discussion that address the  agglutination capability and the difference in gelation of the different matrices generated play a significant role in analyzing the results obtained. This data and discussion belong in the main part of the paper.

Author Response

the answers to the comments and suggestions are in the attached document.

Reviewer 2 Report

The paper describe clearly a complete and consistent amount of pertinent results regarding the investigated systems and its possible application for drugdelivery.

The paer is ceraiinly useful for future technical application but risks to be too descriptive .Some attempts to improve  its scientific value could be done :

The conceptual motivation of the work should be added at the end of the introduction 

2. The starting materials are totally commercial and thier characterization in term of  molecular structure and properties is only accepted and not checked

The possible effect of molecular weight and precise copolymer composition are not reported  and could be important for the  final properties

Molecular interactions between drugs and polymer  as well as drug dispersion are not discussed

Speculation about  possible interpretation at scientific general level  are limited and should be extended

Author Response

the answers to the comments and they are in the attached document
